

# Effects of hydraulic retention time and influent nitrate concentration on solid-phase denitrification system using wheat husk as carbon source

Shuhui Niu[1,2,*], Shuwei Gao[1,2,*], Kai Zhang[1,2], Zhifei Li[1,2], Guangjun Wang[1,2], Hongyan Li[1,2], Yun Xia[1,2], Jingjing Tian[1,2], Ermeng Yu[1,2], Jun Xie[1,2], Minting Zhang[3] and Wangbao Gong[1,2]

[1] Key Laboratory of Tropical and Subtropical Fishery Resource Application and Cultivation, Ministry of Agriculture, Pearl River Fisheries Research Institute, Chinese Academy of Fishery Sciences, Guangzhou, China
[2] Guangdong Ecological Remediation of Aquaculture Pollution Research Center, Guangzhou, China
[3] Guangdong Shunde Junjian Modern Agricultural Technology Co., Ltd, Foshan, China
* These authors contributed equally to this work.

Corresponding author
Wangbao Gong, gongwb@prfri.ac.cn

## ABSTRACT

Solid-phase denitrification shows promise for removing nitrate ($NO_3^-$-N) from water. Biological denitrification uses external carbon sources to remove nitrogen from wastewater, among which agriculture waste is considered the most promising source due to its economic and efficiency advantages. Hydraulic retention time (HRT) and influent nitrate concentration (INC) are the main factors influencing biological denitrification. This study explored the effects of HRT and INC on solid-phase denitrification using wheat husk (WH) as a carbon source. A solid-phase denitrification system with WH carbon source was constructed to explore denitrification performance with differing HRT and INC. The optimal HRT and INC of the wheat husk-denitrification reactor (WH-DR) were 32 h and 50 mg/L, respectively. Under these conditions, $NO_3^-$-N and total nitrogen removal rates were 97.37 ± 2.68% and 94.08 ± 4.01%, respectively. High-throughput sequencing revealed that the dominant phyla in the WH-DR operation were Proteobacteria, Bacteroidetes, and Campilobacterota. Among the dominant genera, *Diaphorobacter* (0.85%), *Ideonella* (0.38%), *Thiobacillus* (4.22%), and *Sulfurifustis* (0.60%) have denitrification functions; *Spirochaeta* (0.47%) is mainly involved in the degradation of WH; and *Acidovorax* (0.37%) and *Azospira* (0.86%) can both denitrify and degrade WH. This study determined the optimal HRT and INC for WH-DR and provides a reference for the development and application of WH as a novel, slow-release carbon source in treating aquaculture wastewater.

## INTRODUCTION

High levels of nitrogen emissions from aquaculture wastewater have become a serious problem worldwide (*Zhu et al., 2015*). Ammonia nitrogen ($NH_4^+$-N) and nitrite nitrogen ($NO_2^-$-N) are toxic to cultured species during aquaculture and are usually converted to nitrate nitrogen ($NO_3^-$-N) by nitrification using biofilters (*Gutierrez-Wing & Malone, 2006*). Nevertheless, this treatment process might not be entirely suitable for aquaculture wastewater, with maximum $NO_3^-$-N levels of 400–500 mg/L in recirculating aquaculture systems (*Liu et al., 2018*).

Heterotrophic biological denitrification is currently one of the most cost-effective methods for $NO_3^-$-N removal from water (*Chen et al., 2015*). Traditionally, liquid organic carbon sources (*e.g.*, methanol, ethanol, and acetic acid) (*Shen, Zhou & Wang, 2013*) or artificial polymers (*e.g.*, polycaprolactone and polybutylene-succinate) (*Luo et al., 2014*; *Zhu et al., 2015*) are applied for denitrification. Nevertheless, the complexity of controlling carbon-source dose and high costs limit their potential applications in related fields (*Luo et al., 2014*). However, natural organic substances (*e.g.*, agricultural waste), which are inexpensive and readily available as solid carbon sources, are promising alternatives for denitrification (*Wang & Chu, 2016*). For example, woodchips (removal efficiency, 60–100%) (*Hoover et al., 2018*), corncobs (removal efficiency, 56–90%) (*Xiong et al., 2020*), and wheat straw (removal efficiency, 75–90%) (*Aslan & Türkman, 2004*) have demonstrated good denitrification performance.

Hydraulic retention time (HRT) is a crucial parameter for bioreactor management (*Cydzik-Kwiatkowska et al., 2014*). Denitrification performance can be improved by increasing the HRT, An optimized HRT ensures efficient hydraulic shear to form denitrifying granular sludge (*Niu et al., 2018*) and sufficient contact time between the substrate and denitrifying bacteria to complete denitrification (*Moussavi, Jafari & Yaghmaeian, 2015*). However, excessively long HRT can reduce denitrification efficiency and lead to nitrite accumulation (*Guo et al., 2017*). Therefore, it is crucial to determine the appropriate HRT for denitrification systems (*Wang, Peng & Stephenson, 2009*). Besides HRT, influent nitrate concentration (INC) significantly affects nutrient removal performance (*Xu et al., 2018*). High INC terminates the denitrification process by inhibiting the production of nitrogen ($N_2$) in favor of $N_2O$, whereas low INC affects the ratio of $N_2O$ to $N_2$ production (*Blackmer & Bremner, 1979*). Previous denitrification systems supported by poly (3-hydroxybutyrate-co-3-hydroxyvalerate) (PHBV) polymer showed different denitrification rates when treating wastewater with different $NO_3^-$-N concentrations (*Xu et al., 2018*). High INC provides more electron acceptors, and the percentage of $NO_3^-$-N removed increases as the amount of $NO_3^-$-N removed increases (*Liu et al., 2017*). Therefore, to ensure the reactor can achieve better $NO_3^-$-N removal, we should understand its maximum treatment limit and make corresponding adjustments to suit the operating conditions.

Wheat, as the most widely cultivated and productive crop worldwide, produces large quantities of by-products, such as wheat husk (WH) (*Barbieri, Lassinantti Gualtieri & Siligardi, 2020*). *Searle & Malins (2013)* estimated that the available wheat residue in 2020

would be 51 million tons in the European Union, among which approximately 20 wt.% would be WH (*i.e.*, approximately 10 million tons). China's wheat output in 2020 was 13,425.4 million tons, and China produces a large amount of WH each year (*National Bureau of Statistics of China, 2021*). Previous studies have focused on WH for the development of bio-based materials (*Barbieri, Lassinantti Gualtieri & Siligardi, 2020*) and its ability to act as a biosorption medium for efficient adsorption of 2,4-dichlorophenol from aqueous solution (*Kalderis et al., 2017*). Our previous research has shown that WH exhibits excellent carbon release capacity and completely removed nitrate and nitrite during a 181-h denitrification reaction (*Gao et al., 2022*). However, less information was revealed on the effects of HRT and INC on solid-phase denitrification system using WH as carbon source in aquaculture wastewater treatment. Hence, the main purpose of this study was to explore optimal HRT and INC parameters for improving nitrogen removal in solid-phase denitrification system using WH as carbon source for treating aquaculture wastewater. In this study, WH was used as an additional carbon source for a one-dimensional denitrification reactor. System performance in removing nitrogen- and phosphorus-containing pollutants was investigated by long-term continuous injection of aquaculture wastewater with different HRTs and INCs using batch experiments. The structural composition and alterations in the bacterial community in the wheat husk-denitrification reactor (WH-DR) were also analyzed using high-throughput sequencing technology. These results provide a theoretical reference and technical guidance for WH as a solid carbon source to enhance denitrification for treating aquaculture wastewater.

## MATERIALS AND METHODS

### Materials preparation

WH, was collected from a household in rural China. The WH was washed twice with deionized water to remove surface dust and other impurities and then dried at 60 °C until its weight was constant (*Xiong et al., 2020*). The WH was then placed into sealed bags and stored until further use. A pond with a culture history was used to collect inoculation denitrification sludge at the Pearl River Fisheries Research Institute in Guangzhou, China. The sludge was filtered through gauze (16 mesh, 1 mm) to remove impurities. The process described by *Li et al. (2019a)* was used to create synthetic aquaculture wastewater (SAT), and its composition is shown in Table S1. The $NO_3^-$-N concentration was adjusted according to different experimental requirements, using $NO_2^-$-N at 2.5 mg/L, $NH_4^+$-N at 5.5 mg/L, and total phosphorus (TP) at 23.88 mg/L. The chemicals (analytical reagents) required for the experiments were purchased from Macklin Biochemical Co., Ltd. (Shanghai, China).

### Setup and operation of WH-DR

As shown in Fig. 1, a cylindrical Plexiglas container was used as a denitrification reactor (DR) (height 10 cm, diameter 55 cm), with a bottom inlet and top automatic overflow located 5 cm above and below the bottom and top, respectively. A towel (100 cm × 40 cm) made of ultrafine chemical fiber material purchased from a supermarket in Guangzhou

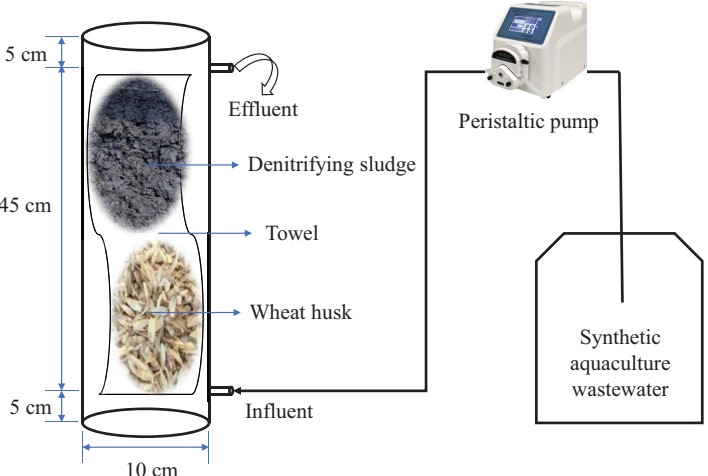

**Figure 1 Schematic diagram of the denitrification system.**

was used as a microbial carrier. It was used to wrap 40 g of WH and 200 mL of inoculation sludge mixture and was placed in the reactor. The effective volume of the reactor was 2,800 ± 100 mL, and the HRT was adjusted by controlling the inlet water flow through a constant flow peristaltic pump (WT-600CAS/353Y type; Huiyu Co., Neijiang, China).

The experiments were conducted in triplicate in a laboratory. WH-DR$_0$ and WH-DR$_{14}$ represent the initial and final stages of the reactor operation (INC = 50 mg/L; HRT = 32 h), respectively.

## Effect of HRT on denitrification performance

Acclimatization was considered successful when $NO_3^-$-N was completely removed, and there was no accumulation of $NO_2^-$-N by filling the WH-DR with SAT (50 mg/L of $NO_3^-$-N) and leaving it to stand. After that, the peristaltic pump was started, and the INC was maintained at 50 mg/L. The WH-DR was operated for 14 d at an HRT of 24, 28, 32, and 36 h. The pH, nitrogen (TN), chemical oxygen demand (COD), dissolved oxygen (DO), and effluent $NO_3^-$-N, $NO_2^-$-N, $NH_4^+$-N concentrations were measured every 2 d.

## Effect of INC on denitrification performance

The acclimatization method was the same as that for the HRT experiment. Based on an HRT of 32 h, the WH-DR was operated at an INC of 75, 100, and 125 mg/L for 14 d. The effluent physicochemical parameters (as above) were measured every 2 d.

## Characterization of physical and chemical properties of WH

WH of WH-DR$_0$ and WH-DR$_{14}$ were collected and dried (60 °C). Scanning electron microscopy (SEM) was used to examine surface structural changes (QUANTA 250; Servicebio Co., Hubei, China). The cellulose, hemicellulose, lignin and carbon, nitrogen, and phosphorus contents of the WH were determined using Microspectrum Detection (Jiangsu WEIPU Testing Technology Co., Ltd, Suzhou, China).

## Microbial community analysis

The denitrifying sludge inside WH-DR$_0$ and WH-DR$_{14}$ was collected for bacterial community diversity analysis. According to the instructions of the E.Z.N.A. soil DNA kit (Omega Bio-Tek, Norcross, GA, USA), approximately 0.3 g of each sludge sample was extracted for DNA analysis. The V3 and V4 highly variable regions of prokaryotic 16S rDNA were amplified using "CCTACGGRRBGCASCAGKVRVGAAT" as the upstream primer and "GGACTACNVGGGTWTCTAATCC" as the downstream primer.
The reaction system used for PCR amplification, included upstream and downstream primers (1 μL of each), dNTPs (2 μL), TransStart Buffer (2.5 μL), TransStart Taq DNA polymerase (0.5 μL), template DNA (20 ng), and ddH$_2$O to a final volume of 25 μL. The construction of high-throughput sequencing libraries and sequencing based on the Illumina MiSeq platform were performed by PANOMIX Biomedical Technology Co., Ltd. (Suzhou, China).

## Analysis methods

DO and pH were determined using a YSI Professional Plus system (YSI Incorporated, Yellow Springs, OH, USA). The concentrations of COD, TN, TP, NO$_3^-$-N, NO$_2^-$-N, and NH$_4^+$-N were determined. Data were collected as previously described in *Zhang et al. (2020)* and *Tu et al. (2010)*. More details on these methods have been described in our previous study (*Gao et al., 2022*). Specifically, NH$_4^+$-N concentrations were measured using the spectrophotometric method with salicylic acid.

WH-DR$_0$ and WH-DR$_{14}$ microbiota were analyzed using STAMP software to identify significant differences in operational taxonomic units (OTUs) and genes. PICRUSt was used to calculate the functional genes of the microbiota based on community structures. All data were analyzed using one-way analysis of variance (ANOVA) and presented as the mean ± standard deviation (SD). Significant differences between the means of different treatments were determined using Duncan's multiple- range tests. Probabilities of $P < 0.05$ were considered significant. GraphPad Prism 8.0.2 (GraphPad Software Inc., San Diego, CA, USA) and Origin 2019b (Origin Lab Inc., Northampton, MA, USA) were used to plot the data.

# RESULTS AND DISCUSSION

## pH and DO changes in WH-DR water

Throughout the experimental period, influent pH was within the 6.70–7.58, whereas effluent pH was slightly lower at 6.43–7.31 (Table 1). In a previous study, solid-phase nitrification using woodchips as a carbon source also exhibited decreased pH (*Zhao et al., 2017*). Theoretically, denitrification consumes acids such as volatile fatty acids from the environment as carbon sources, leading to alkalinity and increased pH (*Elefsiniotis & Li, 2006*). Conversely, the acidic substances such as acetic acid produced during the anaerobic degradation of agricultural waste causes a decrease in pH (*Xu & Chai, 2017*). pH affects the denitrification process mainly by influencing enzyme activity. High denitrification performance requires neutral or near-neutral pH conditions (*Albina et al., 2019*), indicating that the pH conditions in this study were suitable for denitrification. However,

**Table 1 pH and DO changes in WH-DR in each operation stage.**

| Stage | HRT (h) | INC (mg/L) | Influent pH | Effluent pH | Influent DO (mg/L) | Effluent DO (mg/L) |
|-------|---------|------------|-------------|-------------|--------------------|--------------------|
| I | 24 | 50 | 7.30 ± 0.09 | 6.72 ± 0.15** | 3.29 ± 0.19 | 0.36 ± 0.12** |
| II | 28 | 50 | 7.45 ± 0.17 | 7.00 ± 0.06** | 3.21 ± 0.17 | 0.40 ± 0.18** |
| III | 32 | 50 | 7.21 ± 0.13 | 6.87 ± 0.10** | 3.30 ± 0.27 | 0.16 ± 0.06** |
| IV | 36 | 50 | 7.04 ± 0.28 | 6.95 ± 0.05 | 3.20 ± 0.13 | 0.26 ± 0.13** |
| V | 32 | 75 | 7.19 ± 0.05 | 7.13 ± 0.06* | 3.60 ± 0.15 | 0.37 ± 0.11** |
| VI | 32 | 100 | 7.23 ± 0.04 | 7.22 ± 0.08 | 3.41 ± 0.19 | 0.44 ± 0.17** |
| VII | 32 | 125 | 7.14 ± 0.04 | 7.20 ± 0.04* | 3.56 ± 0.15 | 0.54 ± 0.10** |

**Note:**
DO, dissolved oxygen; WH-DR, wheat husk-denitrification reactor; HRT, hydraulic retention time; INC, influent nitrate concentration; "*"represent $p < 0.05$, "**" represent $p < 0.01$

DO was affected in the WH-DR (Table 1). The DO at the reactor outlet declined with increase in HRT. In previous studies, longer HRT was used to reduce DO in the denitrification reactor (*Zhao et al., 2017*; *Xu et al., 2018*). Denitrification occurs mainly under anaerobic conditions, which usually require DO less than 2 mg/L (*Waki et al., 2018*). In this study, the DO in the WH-DR was maintained at a low concentration, and DO at the outlet remained at less than 1 mg/L.

## Effect of HRT on the denitrification performance of WH-DR

HRT is an essential parameter of bioreactor operation. As shown in Fig. 2A, the effluent nitrate concentration (ENC) of the WH-DR was 13.47 ± 3.02 mg/L (HRT of 24 h), 3.53 ± 1.38 mg/L (HRT of 28 h), 1.31 ± 1.34 mg/L (HRT of 32 h), and 0.56 ± 0.49 mg/L (HRT of 36 h) for an INC of 50 mg/L. As show in Fig. 2D, the $NO_3^-$-N removal efficiency was significantly higher at HRTs of 36 and 32 h than at HRTs of 24 and 28 h ($P < 0.05$), indicating that the contact time between $NO_3^-$-N and microbial populations at HRTs of 24 and 28 h was insufficient for $NO_3^-$-N removal (*Moussavi, Jafari & Yaghmaeian, 2015*). In solid-phase denitrification, a long HRT (2.5–24 h) has been used to improve the performance of nitrogen removal performance (*Zhao et al., 2017*; *Xu et al., 2018*). The consumption of organic matter will also be reduced at lower HRTs as it acts as the electron donor for denitrification. Therefore, the effluent COD was relatively high at 52.05 ± 13.52 and 54.91 ± 15.22 mg/L for HRTs of 24 and 28 h, respectively (Fig. 2C). However, if the HRT was relatively long (32–36 h), the treatment efficiency did not increase significantly.

Effluent $NO_2^-$-N and $NO_3^-$-N showed a similar trend (Fig. 2A). At an HRT of 24 h, $NO_2^-$-N accumulated was 1.15 ± 0.68 mg/L, indicating that complete denitrification was not achieved. This is because $NO_3^-$-N reductase would compete with $NO_2^-$-N reductase for substrate electrons and $NO_3^-$-N reductase would inhibit the activity of $NO_2^-$-N reductase (*Ge et al., 2012*). When the HRT was increased to 28, 32, and 36 h, the ENC of WH-DR decreased, the inhibition of $NO_2^-$-N reductase was weakened, and $NO_2^-$-N was completely removed as nitrogen. Adding plant carbon sources may increase the risk of secondary pollution, such as nitrogen release and incomplete nitrogen removal (*Gao et al., 2022*); However, in this study, the removal process of TN and $NO_3^-$-N showed a similar pattern

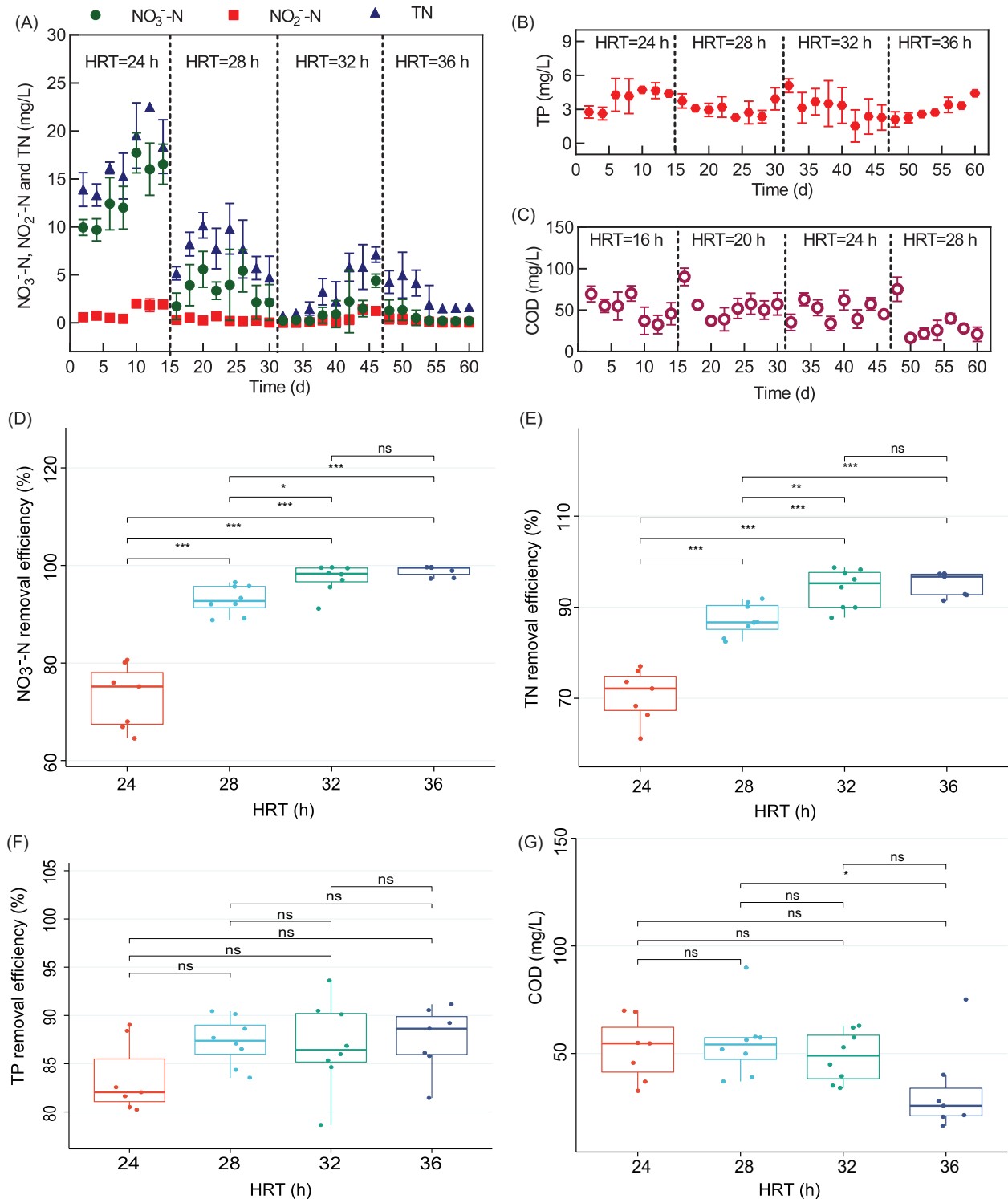

**Figure 2 Removal performance of WH-DR for nitrogen and phosphorus pollutants under different HRTs.** (A) effluent $NO_3^--N$, $NO_2^--N$, and TN concentrations; (B) effluent TP concentration; (C) effluent COD concentration; (D) boxplots of $NO_3^--N$ removal efficiency; (E) boxplots of TN removal efficiency; (F) boxplots of TP removal efficiency; (G) boxplots of COD concentration. WH-DR, wheat husk-denitrification reactor; HRT, hydraulic retention time; TN, total nitrogen; TP, total phosphorous; COD, chemical oxygen demand. Asterisks: $^*p < 0.05$, $^{**}p < 0.01$, $^{***}p < 0.001$.

(Fig. 2). TN removal efficiency was 70.66 ± 5.28% and 87.22 ± 3.30% at HRTs of 24 and 28 h, respectively, and increased significantly to 94.08 ± 4.01% when the HRT was extended to 32 h ($P < 0.05$, Fig. 2E); however, the TN removal efficiency did not increase significantly after reaching a maximum of 95.07 ± 2.43% when the HRT was increased to 36 h. Based on this analysis, we considered 32 h as the optimal HRT of the WH-DR. *Tangsir et al. (2017)* and *Chen et al. (2021)* reported that denitrification reactor could perform efficiently under short HRTs. This result might be caused by the different degradability and structural characteristics of varied materials and feeding types (*Wang & Chu, 2016*). As shown in Fig. 2B, TP removal efficiency gradually increased from 83.49% to 87.56% with longer HRT of 24 to 36 h; however, there were no significant differences among the treatment groups ($P > 0.05$, Fig. 2F), indicating that the effect of HRT on TP removal was not significant between 24 and 36 h, which needs further study.

## Effect of INC on the denitrification performance of WH-DR

The effect of INC on the denitrification performance of the WH-DR was further investigated based on the optimal HRT of 32 h. As shown in Fig. 3A, the ENC was 11.00 ± 3.83, 11.83 ± 2.75, and 24.64 ± 4.96 mg/L when INC was 75, 100, and 125 mg/L, respectively. Nitrate removal rate (NRR) increased significantly when INC was increased to 75, 100, and 125 mg/L ($P < 0.05$, Fig. 3D). These results indicate that INC is the critical factor affecting NRR. The number of electron acceptor per unit volume of the reactor increases with an increase in INC; furthermore, the nitrate volumetric removal rates increase, leading to elevated NRR (*Wang et al., 2020*). *Hoover et al. (2018)* also reported that $NO_3^-$-N removal increased from 7.5 to 12.9 mg/L per unit time in a denitrification system using wood chips as the carbon source when the INC increased from 10 to 50 mg/L. A similar phenomenon was observed for groundwater treatment systems using corncobs as a carbon source (*Liu et al., 2020*).

The effluent COD of the WH-DR decreased with the increase in INC (Fig. 3C). The consumption of organic matter will increase with an increase in $NO_3^-$-N removal as it acts as the electron donor for denitrification (*Gao et al., 2020*). The effluent COD was 52. 24.27 ± 13.99 mg/L for an INC of 75 mg/L, which was significantly higher than that for an INC of 100 mg/L (31.81 ± 9.80 mg/L) and 125 mg/L (28.52 ± 16.41 mg/L) ($P < 0.05$). However, $NO_2^-$-N accumulation gradually increased with the increase in INC (Fig. 3A), indicating that the WH-DR did not achieve complete denitrification. As shown in Fig. 3E, TN removal efficiency decreased gradually with increased INC, at: 82.57 ± 4.22 (INC of 75 mg/L), 81.10 ± 4.88 mg/L (INC of 100 mg/L), and 78.74 ± 4.73% (INC of 125 mg/L), respectively. Combined with the HRT optimization experiment, we observed that the denitrification capacity of the WH-DR saturated at 50 mg/L INC and 32 h HRT. The TP variation curves (Fig. 3B) show no significant differences in TP removal (81.52–83.52%) at INCs of 75–125 mg/L ($P > 0.05$), indicating that INC had a negligible effect on TP removal (Fig. 3F).

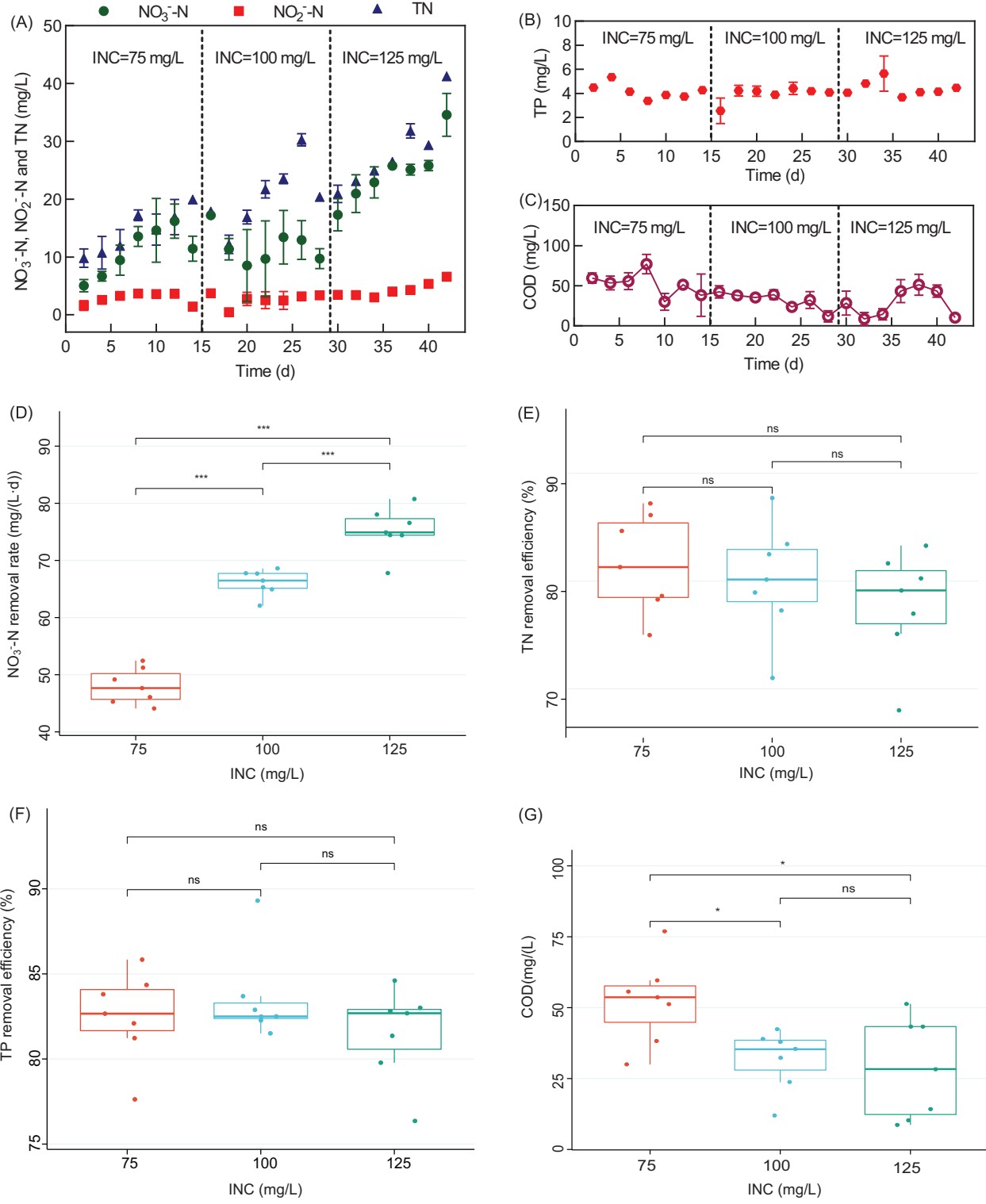

**Figure 3 Performance of the WH-DR for removal of nitrogen and phosphorus pollutants under different INCs.** (A) Effluent $NO_3^--N$, $NO_2^--N$, and TN concentration; (B) effluent TP concentration; (C) effluent COD concentration; (D) boxplots of $NO_3^--N$ removal rate; (E) boxplots of TN removal efficiency; (F) boxplots of TP removal efficiency; (G) boxplots of COD concentration. WH-DR, wheat husk-denitrification reactor; INC, influent nitrate concentration; TN, total nitrogen; TP, total phosphorous; COD, chemical oxygen demand. Asterisks: $^*p < 0.05$, $^{**}p < 0.01$, $^{***}p < 0.001$.

**Table 2 α-diversity indices of the microbial community.**

| Sample | Sequences | OTUs | Coverage | Chao1 | Shannon | Simpson | ACE |
|---|---|---|---|---|---|---|---|
| WH-DR$_0$ | 49,881 | 5,217 | 0.98 | 3,443.61 | 9.07 | 0.99 | 3,519.78 |
| WH-DR$_{14}$ | 44,854 | 4,768 | 0.98 | 3,425.78 | 9.05 | 0.99 | 3,344.43 |

**Note:**

WH-DR$_0$, wheat husk-denitrification reactor at initial stage; WH-DR$_{14}$, wheat husk-denitrification reactor at final stage; OTUs, operational taxonomic units.

## Analysis of the surface structure of WH and changes in the main components

The carbon content of WH can indicate its ability to release carbon, whereas the nitrogen and phosphorus content can reflect its risk of causing secondary pollution. The carbon content of WH was 49.10% (Table S2), which was much higher than the nitrogen and phosphorus content, indicating that WH is less likely to cause secondary pollution.

The main components of agricultural waste include biodegradable cellulose, hemicellulose, and lignin (*Li et al., 2019b*). The extracellular enzymes secreted by attached denitrifying microbial biofilms can degrade cellulose and hemicellulose into soluble small molecule substrates, which can be further utilized by denitrifying bacteria as carbon source in the denitrification process (*Sun et al., 2019*). After 14 d of operation, the cellulose, lignin, and hemicellulose contents of WH decreased, indicating that the microorganisms degraded WH. In addition, the decrease in the carbon content of WH indicated that its carbon release reaction occurred during this period.

The specific surface area and roughness of agricultural waste affect the attachment and growth of microorganisms (*Yang et al., 2015*). SEM images of the original WH at 100× and 500× magnifications are shown in Fig. S1. The WH has a higher specific surface area and roughness because of the dense distribution of conical protrusions on its surface. SEM images of the WH after 14 d at 100× and 500× magnifications are shown in Fig. S1. The original surface structure of the WH was retained; however, its surface attachment substantially increased, implying that a mixture of microorganisms and impurities were attached, indicating that WH is an excellent microbial carrier.

## Microbial community analysis

### Microbial diversity

The α-diversity indices of the microbial communities at the beginning and end of the WH-DR process are shown in Table 2. The Shannon index of WH-DR$_0$ was greater than that of WH-DR$_{14}$, indicating the relatively high community diversity of WH-DR$_0$. In addition, two parameters, Chao1 and ACE indices, were higher for WH-DR$_0$, indicating that it contained more OTUs, *i.e.*, a higher number of species, with greater richness and evenness. Overall, the bacterial community structure of the WH-DR changed after 14 d of operation, and some species were enriched. Similar results were observed following a long operating period of 20 d for a denitrification system that used pig manure as a carbon source (*Luo et al., 2020a*). *Lu, Chandran & Stensel (2014)* suggested that adding

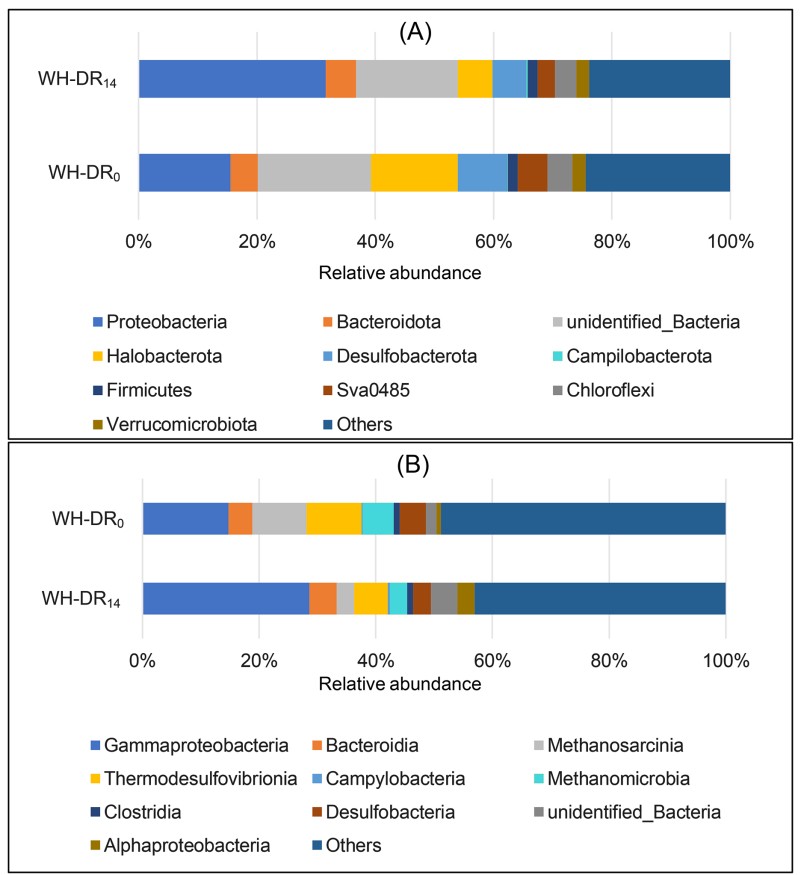

**Figure 4 Relative bacteria abundance of WH-DR$_0$ and WH-DR$_{14}$ at phylum (A) and class (B) levels.** WH-DR$_0$, wheat husk-denitrification reactor at the initial stage; WH-DR$_{14}$, wheat husk-denitrification reactor at the final stage.

agricultural waste and the long-term injection of NO$_3^-$-N were selective for bacteria and altered the bacterial community structure.

## Sludge microbial community during WH-DR operation

The representative sequences of OTUs were classified at the phylum level, as shown in Fig. 4A. The abundance of Proteobacteria increased from 15.53% to 31.61% after 14 d of WH-DR operation. Proteobacteria are the main denitrifying bacteria (*Meng et al., 2017*), and are present in industrial, city (*Ma et al., 2015*), and aquaculture (*Luo et al., 2018*) wastewater treatment systems. The relative abundance of Bacteroidetes, which are also commonly present in the environment and have a vital role in nitrogen cycling and energy conversion in ecosystems (*Zhao et al., 2018*), increased from 4.61% to 5.16%. A higher abundance of Bacteroidetes was also observed in denitrification filter tanks with loofah as a filler (*Zhang, Luan & Du, 2017*). The abundance of Campylobacterota increased from 0.13% to 0.33%. This phylum contains common sulfur autotrophic denitrifying bacteria present in sewage-treatment plants (*Luo et al., 2022*). It is the dominant bacteria in granular anaerobic sludge that can degrade organic matter (*Zhu et al., 2021*). In contrast, the relative abundances of Halobacteria, Desulfobacterota, Firmicutes, Sva0485,

Chloroflexi, and Verrucomicrobiota decreased from 14.68%, 8.35%, 1.69%, 4.99%, 4.19%, and 2.27% to 5.88%, 5.65%, 1.60%, 2.94%, 3.60%, and 2.25%, respectively. Among them, the bacteria involved in denitrification and cellulose degradation were Firmicutes, which were also previously identified in domestic wastewater treatment systems (*Wang et al., 2015*) and play a crucial role in the hydrolysis and acidification of WH. The specific role of Chloroflexi in denitrification has not been reported, but nitrite oxidizing bacteria isolates of Chloroflexi have been reported (*Sorokin et al., 2012*). Bacteria with high lignin degradation capacity in the Verrucomicrobiota phylum (*Qu et al., 2021*) can also promote WH degradation.

As shown in Fig. 4B, the dominant classes in WH-DR$_{14}$ included Gammaproteobacteria (28.65%), Bacteroidia (4.70%), Methanosarcinia (2.91%), Thermodesulfovibrionia (5.83%), Campylobacteria (0.33%), Methanomicrobia (1.06%), Clostridia (1.06%), Desulfobacteria (3.0%), and Alphaproteobacteria (2.95%). The relative abundances of Gammaproteobacteria, Bacteroidia, Campylobacteria, Clostridia, and Alphaproteobacteria increased in WH-DR$_{14}$ compared with those in WH-DR$_0$. Among them, Gammaproteobacteria and Alphaproteobacteria belong to Proteobacteria and contain abundant nitrifying bacteria, anaerobic ammonia oxidizing bacteria, and $NO_2^-$-N oxidizing bacteria, which are the major contributors to nitrogen removal from wastewater systems (*Kumar & Lin, 2010*). The efficient removal of $NH_4^+$-N by WH-DR during this study may be related to the presence of anaerobic ammonia oxidizing bacteria. Gammaproteobacteria and Alphaproteobacteria were the dominant classes in a constructed wetland with wheat straw, cotton, waste newspaper, and poly butylene succinate as carbon sources (*Si et al., 2018*). In addition, as the first dominant class in WH-DR, Gammaproteobacteria also have phosphorus removal functions (*Osaka et al., 2008*). Bacteroidia comprises bacteria with denitrification functions (*Zhang et al., 2019*) and those that decompose macromolecular organics such as protein, cellulose, and lipids (*Cao et al., 2019*), and can promote the removal of nitrate and the decomposition of WH.

Further classification of OTUs at the genus level allowed the observation of more detailed differences in the bacterial community structure between WH-DR$_0$ and WH-DR$_{14}$ (Fig. 5). After 14 d, the dominant genera in the WH-DR changed significantly to *Acidovorax* (0.37%), *Deferrisoma* (0.60%), *Anaeromyxobacter* (0.91%), *Spirochaeta* (0.47%), *Azospirillum* (1.99%), *Azospira* (0.86%), *Diaphorobacter* (0.85%), *Thiobacillus* (4.22%), *Candidatus Nitrotoga* (0.45%), *Ideonella* (0.38%), and *Methanobacterium* (0.54%). Among them, *Acidovorax*, a common genus in solid-phase nitrification systems, could simultaneously degrade organic matter and denitrify. It was reported as the most abundant genus in aquaculture wastewater treatment systems using polycaprolactone (PCL) and PHBV as carbon sources (*Luo et al., 2020b*). *Azospira* could also simultaneously degrade organic matter and denitrify and became the dominant genus after 68 d of operation in a PCL-supported denitrification reactor (*Luo et al., 2018*). *Diaphorobacter* can perform denitrification under both aerobic and anaerobic conditions. It was the most dominant denitrifying bacterium in an industrial sewage treatment system using polylactic acid/PHBV/rice husk composite as the carbon source (*Wang & Chu, 2016*). *Ideonella* is also a denitrifying bacterium. After running a groundwater

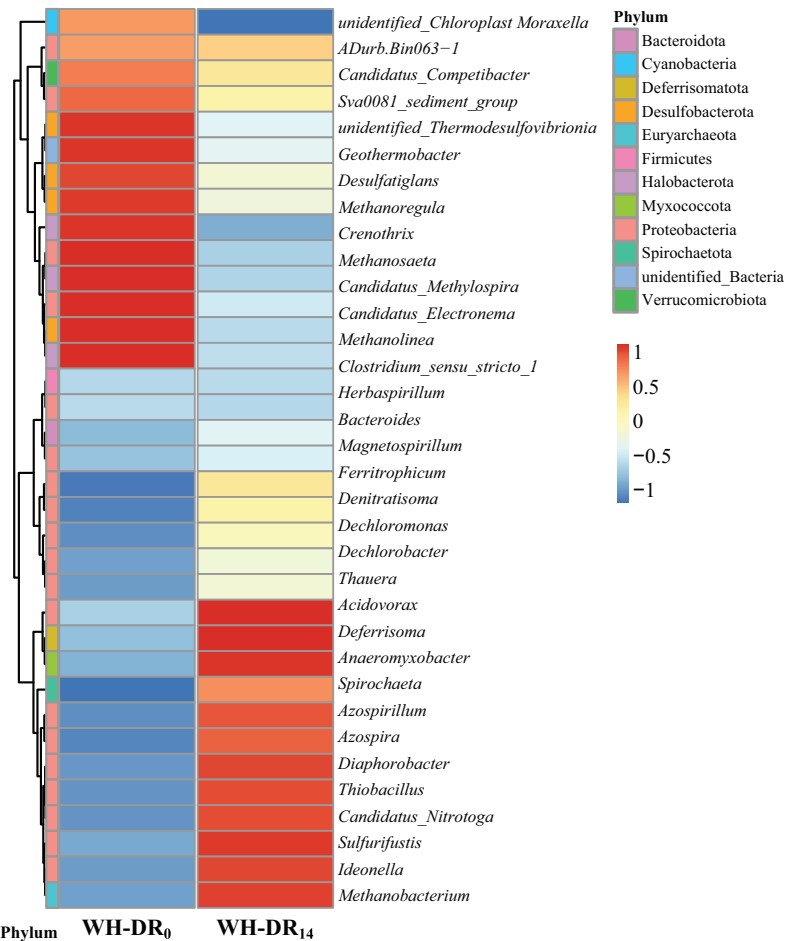

**Figure 5 Heatmap of species abundance clustering of WH-DR$_0$ and WH-DR$_{14}$ based on genus level.** WH-DR$_0$, wheat husk-denitrification reactor at the initial stage; WH-DR$_{14}$, wheat husk-denitrification reactor at the final stage.

denitrification system with rice washing water as the carbon source for a specific duration, *Ideonella* was the dominant bacterium in the inoculated sludge (*He et al., 2019*), which is similar to our findings. *Thiobacillus* is a chemoautotrophic bacterium that can use sulfur as an electron donor for denitrification (*Shi et al., 2022*). Previous studies have found it to be the dominant genus in simultaneous autotrophic and heterotrophic denitrification wastewater treatment systems based on loofah (*Li et al., 2017*). *Candidatus Nitrotoga* is a low-temperature-tolerant nitrifying bacterium (*Skoyles et al., 2020*); its presence in the WH-DR might therefore have contributed to removal of $NH_4^+$-N throughout the experimental period. In a corn cob-supported denitrification system, *Anaeromyxobacter* was mainly involved in dissimilatory nitrate reduction to ammonium (DNRA) (*Sun et al., 2021*); therefore, DNRA may also have occurred in the present study. *Spirochaeta*, a bacterium with hemicellulose-degrading ability, was previously present in a wastewater denitrification system using corn cob as filler (*Zhao et al., 2015*), and its main contribution to this study was the degradation of WH. Collectively, the joint action of these bacteria likely led to the removal of nitrogen pollutants in the WH-DR. In contrast, the relative

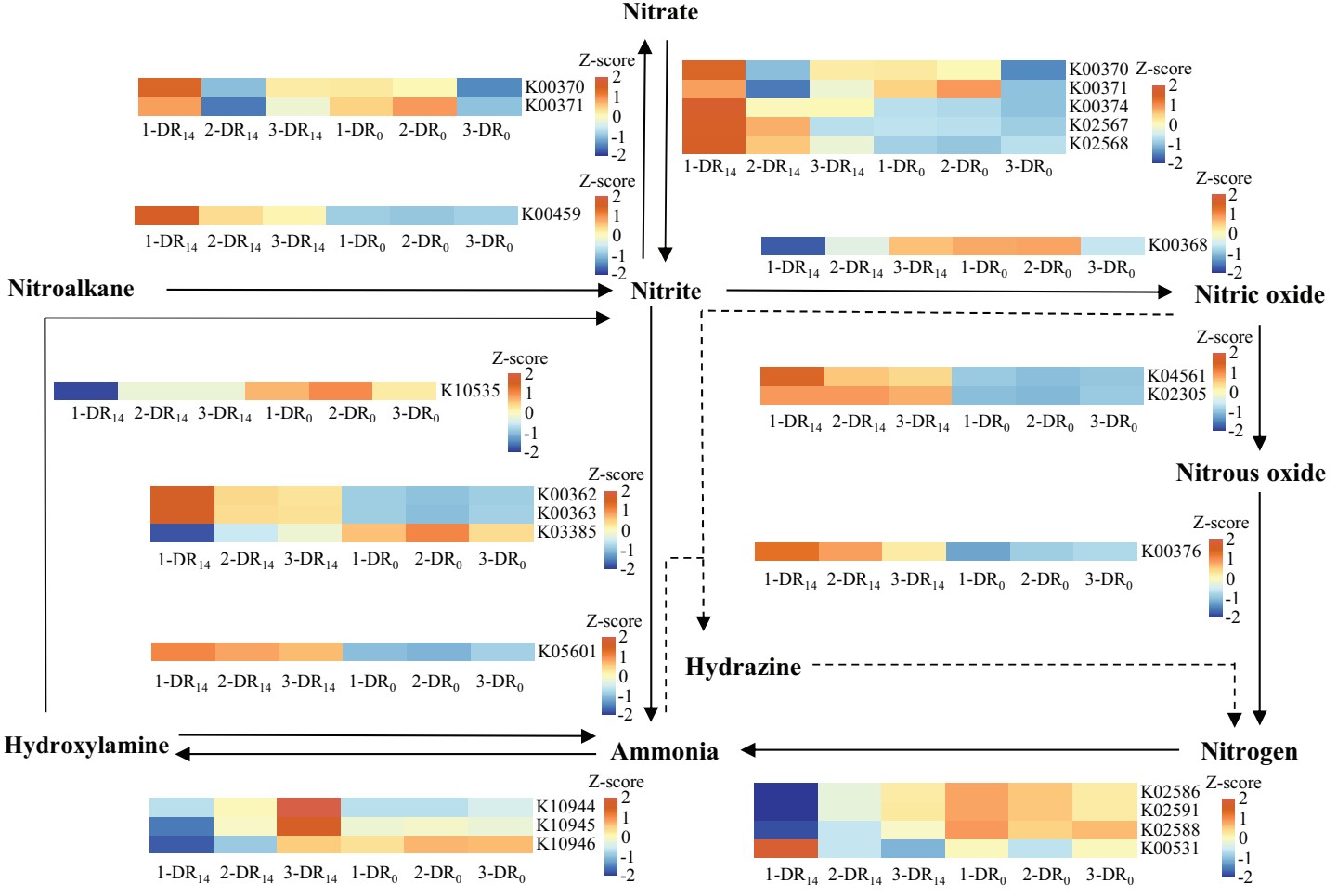

**Figure 6 Changes in the abundance of microbial nitrogen metabolism functional genes at the beginning and end of the WH-DR.** 1-DR$_{14}$, 2-DR$_{14}$ and 3-DR$_{14}$ are at the end of the operation; 1-DR$_0$, 2-DR$_0$ and 3-DR$_0$ are at the beginning of the operation. WH-DR, wheat husk-denitrification reactor.

abundance of some methanogenic archaea such as *Methanosaeta*, *Methanolinea*, *Methanoregula*, and *Crenothrix* decreased within the WH-DR after 14 d of operation, which might be attributed to the increasing organic matter concentration inhibiting their growth metabolism (*Feng et al., 2019*). A similar phenomenon was reported in synchronous denitrification methanogenic systems (*Yi et al., 2016*).

## Changes in the abundance of nitrogen metabolism genes in microorganisms

To verify the metabolic enhancement of biofilm microbiota, their functional genes were determined based on community structure using PICRUSt to analyze genes participating in nitrogen metabolism, as shown in Fig. 6. Overall, the abundance of functional genes related to nitrogen metabolism increased in WH-DR$_{14}$, indicating that microorganisms related to nitrogen metabolism were enriched, which was consistent with the results of the microbial community structure analysis. Among these, genes associated with denitrification (*NarGHI*, *NapAB*, *NirK*, *NorBC*, and *NosZ*) were enriched more

significantly, indicating that $NO_3^-$-N was mainly removed by denitrification. The PICRUSt method provides only a prediction of the metabolic potential within the microbial communities based on the 16S RNA sequencing results. Thus, the functional prediction of microbial communities could be complemented by metagenomics sequencing analysis in future studies.

## CONCLUSIONS

(1) At 50 mg/L INC and 24–36 h HRT, $NO_3^-$-N removal efficiency increased with increasing HRT, but excessively long HRT did not lead to a significant increase in $NO_3^-$-N removal efficiency. The optimal HRT for the WH-DR was 32 h, achieving $NO_3^-$-N removal efficiency of 97.37 ± 2.68%, where $NO_2^-$-N concentration was also low (<0.5 mg/L) and complete denitrification could be achieved. Based on an HRT of 32 h, WH-DR did not achieve complete denitrification when INC was increased to 75, 100, and 125 mg/L, and the NRR of WH-DR increased with INC and the effluent COD decreased with INC. In summary, HRT of 32 h and INC of 50 mg/L provided optimal denitrification performance in the WH-DR.

(2) The bacterial community structure of WH-DR changed after 14 d of operation at an HRT of 32 h and INC of 50 mg/L, in which Proteobacteria, Bacteroidota, and Campilobacterota were enriched. Among the identified dominant genera, *Diaphorobacter*, *Ideonella*, *Thiobacillus*, and *Sulfurifustis* had denitrification function; *Spirochaeta* was mainly involved in the degradation of WH; while *Acidovorax* and *Azospira* were capable of both denitrification and degradation of WH.

Collectively, our findings suggest that the improvement of carbon sources is a promising strategy to increase denitrification performance. Further investigation of the potential of WH as a carbon source and a new approach to treating aquaculture wastewater is warranted. WH showed great potential for denitrification but further study is needed to more specifically characterize carbon release for aquaculture wastewater practice. Moreover, since DO could influence denitrification process, the potential implications of alternatively aerobic/anoxic operational conditions for simultaneous nitrification and denitrification in WH-DR wastewater treatment is an interesting research awaiting future study.

## ACKNOWLEDGEMENTS

We are grateful to the anonymous reviewers for their helpful suggestions. We also thank the anonymous technicians at Servicebio Co., China for assistance with data analysis.

### Funding

This study was funded by the National Key R&D Program of China (2019YFD0900302), the Central Public-interest Scientific Institution Basal Research Fund, CAFS (No.2020TD58), the Fishery Economic Development of Guangdong Province of China (2019B13), and the Modern Agroindustry Technology Research System of China (CARS-

45-21). The funders had no role in study design, data collection and analysis, decision to publish, or preparation of the manuscript.

## Grant Disclosures

The following grant information was disclosed by the authors:
National Key R&D Program of China: 2019YFD0900302.
Central Public-interest Scientific Institution Basal Research Fund: 2020TD58.
Fishery Economic Development of Guangdong Province of China: 2019B13.
Modern Agroindustry Technology Research System of China: CARS-45-21.

## Competing Interests

The authors declare that they have no competing interests.

Minting Zhang is employed by Guangdong Shunde Junjian Modern Agricultural Technology Co., Ltd.

## Author Contributions

- Shuhui Niu conceived and designed the experiments, performed the experiments, analyzed the data, prepared figures and/or tables, authored or reviewed drafts of the article, and approved the final draft.
- Shuwei Gao conceived and designed the experiments, performed the experiments, analyzed the data, prepared figures and/or tables, authored or reviewed drafts of the article, and approved the final draft.
- Kai Zhang performed the experiments, prepared figures and/or tables, and approved the final draft.
- Zhifei Li performed the experiments, prepared figures and/or tables, and approved the final draft.
- Guangjun Wang conceived and designed the experiments, authored or reviewed drafts of the article, and approved the final draft.
- Hongyan Li analyzed the data, prepared figures and/or tables, and approved the final draft.
- Yun Xia analyzed the data, prepared figures and/or tables, and approved the final draft.
- Jingjing Tian analyzed the data, prepared figures and/or tables, and approved the final draft.
- Ermeng Yu analyzed the data, prepared figures and/or tables, and approved the final draft.
- Jun Xie conceived and designed the experiments, authored or reviewed drafts of the article, and approved the final draft.
- Minting Zhang performed the experiments, prepared figures and/or tables, and approved the final draft.
- Wangbao Gong conceived and designed the experiments, performed the experiments, analyzed the data, prepared figures and/or tables, authored or reviewed drafts of the article, and approved the final draft.
## DNA Deposition

The following information was supplied regarding the deposition of DNA sequences:

The assembled genome sequence is available at the National Center for Biotechnology Information (NCBI): PRJNA949132.

## Data Availability

The raw measurements are available in the Supplemental Files.

## Supplemental Information

Supplemental information for this article can be found online at http://dx.doi.org/10.7717/peerj.15756#supplemental-information.

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
