# Peer review of "Effects of hydraulic retention time and influent nitrate concentration on solid-phase denitrification system using wheat husk as carbon source"

_PeerJ, doi:10.7717/peerj.15756_

## Round 0.1 · original submission · Major Revisions

Please incorporate the comments of all reviewers. Additionally, the language of the paper needs attention to improve.

Reviewer 1 ·

Basic reporting

The paper is written nicely. However, following changes are suggested to increase its effectiveness.
Abstract: Direct results are written. It is better to write 2-3 lines, why this study is important to conduct. What methods/material you used for this study.
At the end of Abstract, write a line what is new in your study or what are implications of your results.
Introduction: Introduction is written nicely except last paragraph should be finished at " why and to what extent this study is different from previous literature Effects of hydraulic retention time and influent nitrate
2 concentration on solid-phase denitrification system". Discuss the why this study is necessary to conduct?

Experimental design

Material and Methods is written nicely. Accept my appreciation

Validity of the findings

Results are exactly obtained as methodology adopted. well-done!. My concern is that it is not fully justified based on updated past literature.
Conclusion: This part is written in hurry and is too short, hardly 7 lines. Discuss your own findings based on result obtained. See how your results different/similar with past studies and why and address this. What is novelty and what is left (future research) so that readers can take benefits from your study.

Additional comments

In the nutshell, study is conducted nicely with minor revision. I am waiting the comments to be address suggested by me.

Reviewer 2 ·

Basic reporting

The study investigated the use of wheat husk (WH) as a carbon source to enhance denitrification in a solid reverse nitrification system. Authors evaluated the system operating under different hydraulic retention times (HRTs) and influent nitrate concentrations (INCs). Wheat husk as a waste from agriculture, recycling them in the denitrification will contribute to the development of sustainable agriculture.
The experiment design in this study is simple, but abundant data was collected. English language is OK, though typos can be found. The authors should proofread the manuscript carefully to catch any other errors. Overall, this is a well-conducted study with meaningful findings. My questions focused on the materials and methods and the presentation of the results, as can be found in my following comments.

Experimental design

1. Line 57-60: Please be consistent in the name of nitrogen. Since you explained the abbreviation of ammonia nitrogen, you should also explain nitrite nitrogen and nitrate nitrogen. In line 60, please revise ‘low NH4+-N level and high nitrate accumulation’. Please also check the whole manuscript for similar problems.
2. Line 113: What is the cleaning procedure?
3. Please explain the abbreviation of dissolved oxygen in materials and methods (at first appearance).
4. There are some typos in the figure 5 caption.
5. Authors introduced there were 3 replicates in this experiment. Therefore, in Table 1, instead of presenting the range of the results, it is preferred to present the mean and standard deviation/error and do mean separations as in other figures/tables. Besides, ‘phase’ is confusing here. I would expect from phase 1 to phase 7 to be a continuous period. However, they were not.

Validity of the findings

no comment

---

## Round 0.2 · accepted · Accept

The paper has been improved after revisions and is accepted for publication. Thanks for your fine contribution to PeerJ.

Reviewer 1 ·

Basic reporting

No comments

Experimental design

No comment

Validity of the findings

No comment

Additional comments

Paper is ready for publication in my opinion. Authors truely incorporated what I inquired.

Reviewer 2 ·

Basic reporting

I appreciate the authors' careful revision. My comments have been fully addressed, and I have no further comments. I recommend that this manuscript be accepted for publication.

Experimental design

no comment

Validity of the findings

no comment